# Efficacy Evaluation of Two Commercial Vaccines Against a Recombinant PRRSV2 Strain ZJnb16-2 From Lineage 8 and 3 in China

**DOI:** 10.3390/pathogens9010059

**Published:** 2020-01-15

**Authors:** Guangwei Han, Huiling Xu, Yanli Wang, Zehui Liu, Fang He

**Affiliations:** Institute of Preventive Veterinary Medicine, College of Animal Sciences of Zhejiang University, Hangzhou 310058, China; 11617032@zju.edu.cn (G.H.);

**Keywords:** porcine reproductive and respiratory syndrome virus (PRRSV), modified-live virus vaccines, efficacy evaluation, recombination

## Abstract

From 2010, novel recombinant lineage 3 of porcine reproductive and respiratory syndrome virus 2 (PRRSV2) has continuously emerged China, which has brought about clinical outbreaks of the disease. Previously, a PRRSV2 strain named ZJnb16-2 was identified as a recombinant virus from lineage 8 and 3. In this study, two modified-live vaccines VR2332 MLV and HuN4-F112, which belong to lineage 5 and 8 respectively, were used for efficacy evaluation against the challenge of ZJnb16-2. Piglets vaccinated with HuN4-F112 exhibited temporary fever, higher average daily weight gain, and mild clinical signs as compared to VR2332 MLV vaccinated and unvaccinated piglets upon ZJnb16-2 challenge. Both vaccines could inhibit virus replication in piglets at 21days post challenge (DPC). Cross-reactivity of interferon (IFN)-γ secreting cells against ZJnb16-2 were detected in both vaccinated piglets. The number of IFN-γ secreting cells against ZJnb16-2 in the vaccination group exhibited sustaining elevation after challenge. Results demonstrated that both vaccines provided partial protection against ZJnb16-2 infection. A cross-neutralization antibody against ZJnb16-2 was not detected in any vaccinated piglet before challenge. A low neutralizing antibody titer against ZJnb16-2 was detected after challenge. Besides, all the vaccinated piglets suffered from different degrees of lung pathological lesions, indicating neither VR2332 MLV nor HuN4-F112 provided full protection against ZJnb16-2. This study provides valuable guidelines to control the recombinant virus from lineage 8 and 3 infection with MLV vaccines in the field.

## 1. Introduction

Porcine reproductive and respiratory syndrome (PRRS) are caused by the porcine reproductive and respiratory syndrome virus. Porcine reproductive and respiratory syndrome virus (PRRSV) leads to reproductive disturbance in sow herd and respiratory disease in growing pigs [1,2]. PRRSV from the Arteriviridae family is a single-stranded positive-sense RNA virus which is characterized by extensively genetic variation [3,4]. The genome of PRRSV is about 15 kb which encodes at least 11 open reading frames (ORFs) (1a, 1b, 2a, 2b, 3, 4, 5a, 5, 6, 7, and a short transframe ORF) that are expressed from genomic and subgenomic (sg) mRNAs (sgmRNAs) [5]. PRRSV includes two kinds of viruses, PRRSV1 and PRRSV2 [5]. PRRSV1 and PRRSV2 together with equine arteritis virus (EAV), lactate dehydrogenase-elevating virus (LDV) and simian hemorrhagic fever virus (SHFV) are classified to family Arteriviridae [6].

PRRSV2 is further divided into 9 lineages based on phylogenetic analysis of ORF5 [7]. The first PRRSV2 strain CH-1a of China was isolated in 1996 in lineage 8 [8]. In 2006, a highly pathogenic PRRSV (HP-PRRSV) strain (lineage 8) with a 30-amino acid deletion in its NSP2 protein emerged and became the dominant PRRSV2 strain for the following years in China [9]. From 2013, novel NADC30-like PRRSV strains (lineage 1) with 131-aa discontinuous deletions in the nonstructural protein 2 (nsp2) have been isolated in several provinces in China [10,11]. Lineage 3 strains were initially reported in Taiwan and have emerged in Hong Kong in 2004 based on phylogeography and phylodynamics analysis [12,13]. The representative isolate of lineage 3 was QYYZ reported in 2010 and QYYZ-like strain named GM2 were isolated from mainland China in 2011, which were confirmed to be recombinant strains among QYYZ and VR2332 MLV [14]. The emergence of different PRRSV subgroups made the epidemic situation of PRRSV more complicated, suggesting that current commercial vaccines may not provide sufficient protection with the risks in new outbreaks of PRRSV2. 

Recombination and extensive genetic mutation increased the genetic diversity of PRRSV2 in China. The representative isolate of lineage 3 is QYYZ and QYYZ shares 85.7%, 86.7%, 88.1%, and 80.6% nucleotide identities with VR2332, JXA1, CH-1a, and NADC30, respectively. Some recombinant strains between QYYZ-like and HP-PRRSV have been documented, and lineage 3 PRRSV has become epidemic since 2013 [14,15,16,17,18,19,20]. The recombination area provided by a minor parent was listed. Based on recombination and full-length genomic sequence analysis, the ORF2-ORF7 segment of most isolates belong to lineage 3 and the remaining segments are closely related to the Chinese HP-PRRSV. Whether great changes of amino acid sequence in structural proteins affect vaccine efficacy remains a question to be answered. 

The typical clinical signs of piglets caused by PRRSV2 include pyrexia, dyspnea and irregular respiration, and average daily weight gain (ADWG) reduction. Seven commercially modified-live virus vaccines of different lineages are widely used to control PRRS in China including VR2332 (lineage 5), R98 (lineage 5), JXA1-P80 (lineage 8), HuN4-F112 (lineage 8), GDr180 (lineage 8), TJM-F92 (lineage 8), and CH-1R (lineage 8). Many studies confirmed that attenuated virus vaccines can provide protection against homologous isolates and have positive effects in improving clinical performance and relieving lung lesions and viremia [21,22,23]. However, only partial protection could be provided against newly emerging heterologous PRRSV strains by those vaccines [24,25,26]. In phylogenetic analysis, ZJnb16-2 was identified as a recombinant virus from lineage 8 and 3 [15]. Phylogenetic analysis results indicated that ZJnb16-2 and other similar PRRSVs (GD1404, SCcd16, Scya17, SH-CH, XJzx1-2015) formed a new inter-subgenotype among lineage 8 based on nucleotide sequences of the full-length genome [15,27]. Currently, protection evaluation has not been reported of conventional modified-live vaccines against these newly emerging viruses. In this study, efficacy evaluation of two modified-live virus vaccines against ZJnb16-2 was conducted.

## 2. Results

### 2.1. Clinical Signs in Vaccinated Piglets upon ZJnb16-2 Challenge

All piglets were clinically normal according to respiratory scores and rectal temperatures after vaccination confirming the safety of two commercial PRRSV vaccines. As shown in Figure 1A, the VR2332 MLV/ZJnb16-2 group and the Mock/ZJnb16-2 group developed a fever (above 40 °C) at 2 days post challenge (DPC) which lasted for consecutive 9 days. Piglets in the HuN4-F112/ZJnb16-2 group exhibited a mild fever at 4–8 DPC (40 °C). The rectal temperature of piglets in the healthy control group was normal and ranged from 38.5 °C to 39.5 °C during the whole experiment. The mean score of clinical signs was significantly lower in the HuN4-F112/ZJnb16-2 group compared to the groups VR2332 MLV/ZJnb16-2 and Mock/ZJnb16-2 from 4DPC to 12 DPC (*p* < 0.05; Figure 1B. ADWG is an important indicator of the growth performance of the challenged pigs. As shown in Figure 1C, ADWG in the HuN4-F112/ZJnb16-2 and healthy groups were significantly higher than the VR2332 MLV/ZJnb16-2 and Dulbecco’s Modified Eagle Media (DMEM)/ZJnb16-2 groups at 14 DPC and 21 DPC (*p* < 0.05). One piglet died at 14 DPC in the MLV/ZJnb16-2 group. The cause of death was determined as PRRSV using RT-PCR.

### 2.2. Viremia Test

Viremia was only detectable in one piglet from the HuN4-F112/ZJnb16-2 group before challenge (Figure 2A). At 7 and 14 DPC, the virus titer in piglet sera of the VR2332 MLV/ZJnb16-2 group was similar to the DMEM/ZJnb16-2 group (Figure 2B,C). The viral loads of piglets in the HuN4-F112/ZJnb16-2 group was significantly lower than that of the DMEM/ZJnb16-2 group at 14 DPC (*p* < 0.05; Figure 2C). The viral loads in all vaccinated piglets were lower than that of the DMEM/ZJnb16-2 group at 21 DPC (*p* < 0.05; Figure 2D). All piglets in the control group were PRRSV negative in the study.

### 2.3. Serological Test

As shown in Figure 3A, all vaccinated piglets became seroconverted at 14 days post vaccination (DPV). These modified-live virus vaccines were able to induce an antibody response against PRRSV at the similar levels (Figure 3A). At 28 DPV, a low titer of homologous neutralizing antibody could be detected (≤1:8) in the VR2332 MLV/ZJnb16-2 and HuN4-F112/ZJnb16-2 groups respectively (Figure 3B,C). A neutralization antibody titer against homologous vaccine strains was 1:16 on average at 21 DPC. No cross-neutralization reactivity against ZJnb16-2 was detected in either vaccinated group before challenge (Figure 3D). Neutralizing antibodies against ZJnb16-2 were detected after challenge and the titer was 1:4 at 21 DPC (Figure 3D).

### 2.4. Cross-Reactivity of Interferon (IFN)-γ SC Against ZJnb16-2 was Detected in Both Vaccinated Piglets 

Piglets from both vaccinated groups had (*p* < 0.05) higher numbers of each PRRSV-strain-specific IFN-γ-SC (HuN4-F112, VR2332 MLV and ZJnb16-2) in peripheral blood mononuclear cell (PBMC)compared to the unvaccinated and unchallenged groups (Figure 4A–C). At 21 and 28 DPV, piglets vaccinated with HuN4-F112 produced more ZJnb16-2 IFN-γ-SCs than the VR2332 MLV/ZJnb16-2 group and the statistical difference was not significant (Figure 4C). The number of IFN-γ secreting cells against ZJnb16-2 in the vaccinated group exhibited sustaining elevation after challenge (Figure 4C).

### 2.5. Pathological and Histopathological Examination

Overall, piglets from the HuN4-F112/ZJnb16-2 group had significantly lower scores for macroscopic lung lesions than VR2332 MLV/ZJnb16-2 group (*p* < 0.01; Figure 5A). Lung pathological lesions including multifocal and tan-mottled areas with irregular and indistinct borders could be observed in both ZJnb16-2 challenged piglets after necropsy (Figure 5B). All challenged piglets in the HuN4-F112/ZJnb16-2 group exhibited mild gross lesions, and one presented enlargement of lymph nodes (Figure 5B). The dead piglets in the VR2332MLV/ZJnb16-2 group and all piglets in DMEM/ZJnb16-2 group exhibited severe pathological lung lesions including pulmonary edema and congestion. Enlargement and hyperemia of the lymph nodes were observed in the VR2332 MLV/ZJnb16-2 and DMEM/ZJnb16-2 groups. The VR2332 MLV/ZJnb16-2 group presented a similar level of pathological lesions to the unprotected Mock/ZJnb16-2 group. 

Collapsed alveoli with infiltration of numerous inflammatory cells in alveolar spaces and exfoliated epithelial cells in the bronchiole could be observed in all challenged piglets under microscopic examination (Figure 6). Enlarged lymph nodes in the challenged group exhibited less proliferation of cortical lymphocytes, hyperplasia of the lymphatic nodules, and scattered distribution of cellular debris. No macroscopic and histological changes in the lung and lymph nodes were observed in piglets of healthy controls in the experiment.

9D9, a mAb against N protein, was used for PRRSV antigen detection in lungs by immunohistochemistry (IHC). Results revealed that different intensities of positive staining in lung and lymph nodes were detected in all piglets except the control group (Figure 6). No positive signals were detected in the control group.

## 3. Discussion

Two attenuated PRRSV vaccine strains VR2332 MLV and HuN4-F112 were used to evaluate the efficacy against the ZJnb16-2 infection. Results indicated that both two vaccines provided partial protection upon ZJnb16-2 challenge, and HuN4-F112 seemed to be more effective in controlling the negative consequences of challenge. From the clinical manifestations, the VR2332 MLV/ZJnb16-2 and DMEM/ZJnb16 piglets exhibited severe clinical signs from 3 days post challenge, including prolonged high fever (40.5 °C), inappetence, lethargic, rubefaction, tachypnea, dyspnea and ataxia. One piglet died from the MLV/ZJnb16-2 group at 14 DPC. However, only a short duration of high fever (40 °C) and mild clinical signs were observed in piglets of the HuN4-F112/ZJnb16-2 group. In addition, VR2332 MLV vaccinated piglets showed lower ADWG at 7 DPC, 14DPC and 21 DPC, while the HuN4-F112 vaccinated piglets had a similar ADWG to the unchallenged control group at 14 DPC and 21 DPC. Results of virus titer determination in serum indicated that both vaccines partially reduced the level of viremia in piglets at 21 DPC. In contrast, virus titer was 10^2.5^/ml on average in the DMEM/ZJnb16-2 group at 21 DPC. Virus titer of both vaccinated groups was similar to unvaccinated/unchallenged piglets at 7 and 14 DPC. Pathological analyses indicated that vaccination of HuN4-F112 could reduce gross and microscopic lung lesions of piglets upon ZJnb16-2 challenge compared to the DMEM/ZJnb16-2 group. In contrast, similar levels of gross and microscopic lung lesions could be observed in the VR2332 MLV/ZJnb16-2 and DMEM/ZJnb16-2 groups. 

A neutralizing antibody plays an important role in the control of PRRSV infection [28]. Many linear neutralizing epitopes of PRRSV have been identified [29,30,31]. The exposure to multiple heterologous strains over time is considered to trigger the appearance of broadly neutralizing antibodies [32]. In this study, a neutralizing antibody could only be detected against the homologous vaccine strain in vaccinated piglets at 28 DPV, and the average neutralizing antibody titer was about 1:8. A neutralizing antibody against ZJnb16-2 was not detectable in any piglet at 28 DPV. This phenomenon may be caused by extensive mutation and low genetic identity of structural proteins. A neutralization antibody against ZJnb16-2 could be detected in vaccinated piglets at 21 DPC. Though the neutralizing antibody titer against ZJnb16-2 was low at 21 DPC, neutralizing antibodies may play a role in viremia reduction at 21 DPC in both vaccination piglets. Besides, cell-mediated immunity (CMI) is another crucial factor to control PRRSV infection [33,34]. IFN-γ is an important cytokine that could enhance activity of NK cells and promote antigen presentation. Broadly cross-reactivity of cell-mediated immunity could be detected among type 2 PRRSVs [35]. ORF1a and ORF1b of ZJnb16-2 shared 95.4% and 97.9% identities in amino acids sequence with HuN4-F112, indicating many T-cell epitopes were conservative among them [36,37,38]. In this study, IFN-γ secreting cells against ZJnb16-2 in both vaccinated groups were detected confirming the existence of cross-reactivity of IFN-γSC. Piglet vaccination of HuN4-F112 produced a higher number of IFN-γ secreting cells than VR2332 at 2 8DPV, 7 DPC, and 14 DPC. This may explain why HuN4-F112 is more effective in controlling the negative effects upon challenge. The presence of IFN-γ secreting cells against ZJnb16-2 in both vaccinated groups may contribute to the viremia reduction at 21 DPC. 

Vaccination with modified-live virus vaccines is the most common strategy used to control the spread of PRRS. However, the safety and cross-protective efficacy against heterologous strains of modified-live virus vaccines should be carefully evaluated before they are used in the field. However, studies have indicated that currently commercial available PRRS vaccines only provide partial protection against heterologous strains of PRRSV [24,26,39]. The obstacles of vaccines in controlling PRRSV include viral suppression of innate immunity and the delayed adaptive response of the host resulting in the failure of clearance of heterologous viruses [40]. Thus a replication-competent and IFN antagonism-negative PRRSV vaccine will improve the host anti-viral immune response including the innate immunity, CD4+ T cells, CTLs, and B cells, and provide effective protection not only from homologous infection but also from heterologous infection [41]. In addition, single cell technologies used for generating broadly neutralizing anti-PRRSV antibodies could be an effective strategy for PRRSV control [42]. 

Overall, neither modified-live virus vaccines tested could provide complete protection against the heterologous ZJnb16-2 challenge based on clinical signs, viremia, neutralization antibody levels, and pathological examination. These findings highlighted the necessity for new vaccine strategies which could provide sufficient protection against different PRRSVs. Cross-protection of modified attenuated vaccines against lineage 3 PRRSV with no recombination signals should also be further studied. 

## 4. Materials and Methods

### 4.1. Cell, Vaccines, and Virus

Porcine Alveolar Macrophage (PAMs) were prepared as described previously for PRRSV ZJnb16-2 growth, virus titer determination, and cross-neutralization reactivity assessment [43]. Two commercially modified-live PRRS vaccines with their corresponding virus strains named VR-2332 (lineage 5; Ingelvac PRRS® MLV, Ingelheim, Germany) and HuN4-F112 (lineage 8; Harbin Weike Biotechnology Development Company, Harbin, China) were used in this study. The JXA1-like and QYYZ-like recombinant strain ZJnb16-2 (lineage 8, 3) was previously isolated in 2016 from Zhejiang, China.

### 4.2. Animal Experiments and Clinical Assessment

Twenty-four 3-week-old piglets were purchased from farms located in remote and isolated rural areas. Piglets were not immunized with any vaccine before the study. Piglets were confirmed to be free of reproductive and respiratory syndrome virus (PRRSV), classical swine fever virus (CSFV), pseudorabies virus (PRV), and porcine circovirus 2 (PCV2) before the experiment. There was no PRRSV, CSFV, PRV, or PCV2 specific antibody detected using commercial ELISA kits before the study. All piglets were randomly divided into four groups (*n* = 6) including the HuN4-F112 vaccinated/ZJnb16-2 challenged group, VR2332MLV vaccinated/ZJnb16-2 challenged group, Mock/ZJnb16-2 challenged group, and Control group. Piglets within each group were housed separately. Animal experiments were performed in Zhengli Antoo Biotech of Zhejiang, China. Immunization was performed at 5 weeks of age. Piglets in the HuN4-F112/ZJnb16-2 and VR2332 MLV/ZJnb16-2 groups were intramuscularly vaccinated with corresponding commercial PRRSV modified-live vaccines under manufacturers’ recommendations. Piglets in the DMEM/ZJnb16-2 group received the same volume of DMEM medium. At 28 days post vaccination (DPC), all groups except the negative control group were intranasally challenged with 3 mL of ZJnb16-2 (10^4.5^TCID_50_/m). Animal experiments were approved by the Animal Welfare and Ethics Committee at Laboratory animal center of Zhejiang University (Approve Number 12095). All experiments were performed in accordance with relevant guidelines and regulations. After viral challenge, the rectal temperature and clinical signs of piglets were monitored and recorded daily using a scoring system as previously described [44]. Briefly, clinical signs scoring system included gross clinical score (GCS), respiratory clinical score (RCS), and nervous signs score (NSS). The sum of GCS, RCS, and NSS was the total score of each piglet. Each piglet was scored daily after challenge and the mean score of 1–3 DPC, 4–6 DPC, 7–9DPC, 10–12DPC, 13–15DPC, 16–18DPC, and 19–21 DPCwere counted. Body weight measurements were performed at 0, 7, 14, and 21 DPC in order to calculate ADWG. All surviving piglets were humanely euthanized 21 days after viral challenge.

### 4.3. Viremia and Virus Titration

The viral titers of PRRSV were determined by using a microtitration infectivity assay and expressed as 50% tissue culture infective dose per mL (TCID_50/_mL). Serum samples were collected at 0, 7, 14, 21, and 28 DPV and 7, 14, and 21 DPC. Virus titration was checked in MARC-145 cells before challenge and viremia was determined in PAM cells after challenge. All samples were serial diluted to determine 50% tissue culture infective dose (TCID_50_). Briefly, the monolayer Marc-145 cells (10^5^ cell/well) and PAM cells cultured in 96-well plates (10^5^ cell/well) were incubated with 10-fold serially diluted serum samples at 37 °C for 1 h. Supernatants were removed and replaced with 100 ul DMEM (2% fetal bovine serum) and RPMI (Roswell Park Memorial Institute)-1640 containing 10% fetal bovine serum and 1% antibiotic–antimycotic (10,000 units/mL of penicillin, 10,000 µg/mL of streptomycin, and 25 µg/mL of Amphotericin B). Plates were incubated under a humidified 5% CO_2_ atmosphere at 37 °C for 36 h, and then virus titers were determined in IFA using PRRSV specific monoclonal antibody 9D9. The titers were calculated using Reed-Muench method [45].

### 4.4. Assessment of PRRSV Antibody 

The PRRSV N antibody in serum was measured using a commercial IDEXX Herdchek PRRS 2XR ELISA kit (Westbrook, ME, USA) according to the manufacturer’s directions. The samples were considered to be positive for PRRSV if the sample-to-positive (S/P) ratio was equal to or higher than 0.4. 

The virus neutralization titer against vaccine strains and ZJnb16-2 was performed in Marc-145 cells ((10^5^ cell/well) and PAM cells (10^5^ cell/well) in 96-well plates. Sera were diluted using a two-fold serial dilution technique with RPMI-1640 medium and incubated with 100 TCID_50_ of virus at 37 °C for 1 h. Mixtures were transferred into cells in 96-well plates and incubated at 37 °C for 36 h. Each dilution was repeated in 4 wells. The plates were fixed in 80% acetone and the infection in each well was determined in IFA with mAb 9D9 against the PRRSV N protein. The neutralizing antibody titer was calculated as the last dilution that showed a 90% reduction in the number of fluorescent foci [46]. 

### 4.5. PRRSV-Specific IFN-γ ELISpot 

Peripheral blood mononuclear cell (PBMC) were isolated by density gradient centrifugation using a commercial kit (Solarbio Science & Technology Company, Beijing, China) according to the manufacturer’s directions. PRRSV-specific T-lymphocyte response was evaluated using a commercial product (Mabtech, Cincinnati, OH, USA) as manufacturer’s instructions. Briefly, pIFNγ-I monoclonal pre-coated plates were incubated with RPMI containing 10% serum (200 uL/well) for 30 min at room temperature. The medium was then removed, and the plate was washed with PBS 5 times (200 uL/well). Each well was added with 5 × 10^5^ PBMC and stimulated with 5 × 10^4^ TCID50 of PRRSV at 37 °C with 5% CO_2_ for 48 h. Cells were removed and the plates were washed 5 times with PBS (200 uL/well). Each well was sequentially incubated with 100 uL of biotinylated mAb P2C11 (0.5 ug/mL) for 2 h and streptavidin-ALP (1:1000 in PBS with 0.5% FCS) for 1 h at room temperature. Finally, filtered BCIP/NBT-plus was added to develop spots. Spots were counted and analyzed using a Cellular Technology Limited (CTL) ELISpot analysis system (Cleveland, USA).

### 4.6. Lung Gross Pathological and Microscopic Lesion Examination

All the surviving piglets were humanly euthanized and autopsied at 21 DPC. A scoring system for gross lung lesions was used as previously described [47]. Lung and lymph node tissues were collected and fixed in 4% paraformaldehyde solution for hematoxylin and eosin (H&E) and immunohistochemistry staining. The mAb 9D9 against PRRSV-N was used for immunohistochemistry staining diluted at 1:50.

RT-PCR and PCR method were used to confirm the cause of the dead piglets in the VR2332 MLV group. Conservative primers based PRRSV ORF5 and NSP2, PCV2 ORF2 were designed to identify pathogens in the lungs. In addition, a universal primer aimed at 16sRNA was designed to exclude the cause of mycoplasmal pneumonia of swine.

### 4.7. Statistical Analysis

The statistic difference of rectal temperature, clinical score, average daily gain between the mock group and challenged group was analyzed using *t*-test. A *t*-test was used to analyze the significant of differences of viremia and gross lung lesions between the DMEM/ZJnb16-2 group and two vaccination groups. The *p*-values <0.05 were considered statistically significant. The data are expressed as the mean ± standard error of the mean. The number of piglets in the VR2332 MLV/ZJnb16-2 group was five in the last 14 days after challenge in the process of data analysis and processing. Except that, the number of piglets in each treatment group was six.

## Figures and Tables

**Figure 1 pathogens-09-00059-f001:**
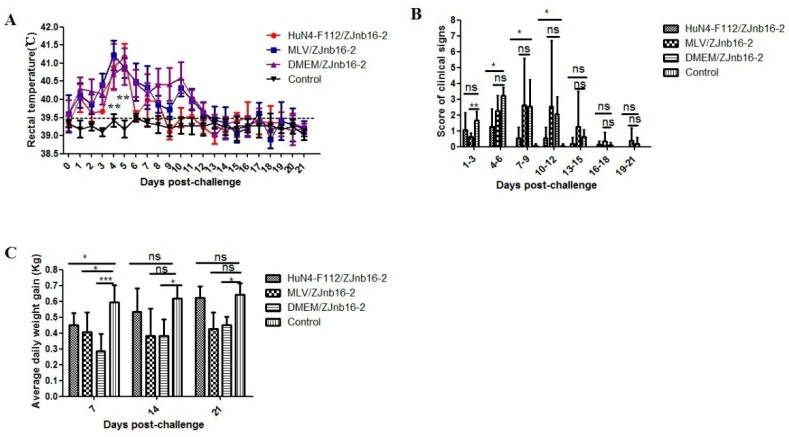
Rectal temperature, clinical sign scores, and average daily weight gain of piglets after ZJnb16-2 challenge. (**A**) Rectal temperatures shown are mean ± standard error (error bars); (**B**) The daily scores of clinical signs; (**C**) Average daily weight gain (ADWG) shown are mean ± standard error (error bars). Asterisk indicates significant differences between all challenged groups and healthy control group (* *p* <0.05; ** *p* < 0.01; *** *p* < 0.001, ns means no significant differences).

**Figure 2 pathogens-09-00059-f002:**
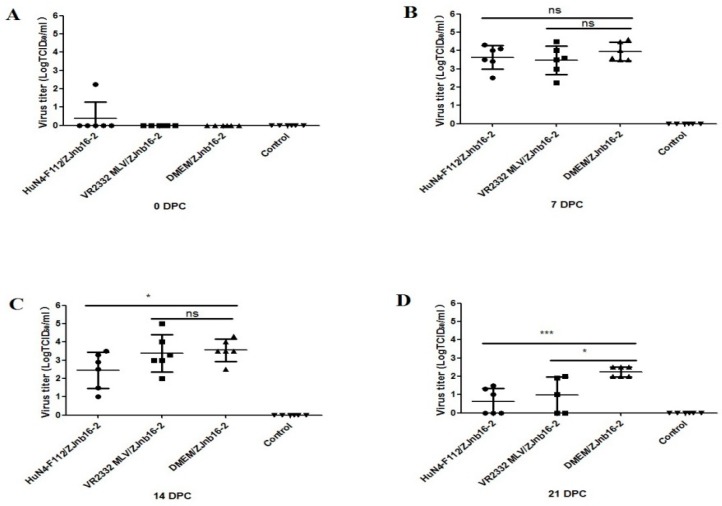
Virus titers in sera of the challenged piglets at 0, 7, 14, and 21 days post challenge (DPC) (**A**–**D**). Data were shown as means ± standard error (error bars). Asterisk indicates significant differences between vaccinated groups and Dulbecco’s Modified Eagle Media (DMEM) group (* *p* < 0.05; *** *p* < 0.001, ns means no significant differences).

**Figure 3 pathogens-09-00059-f003:**
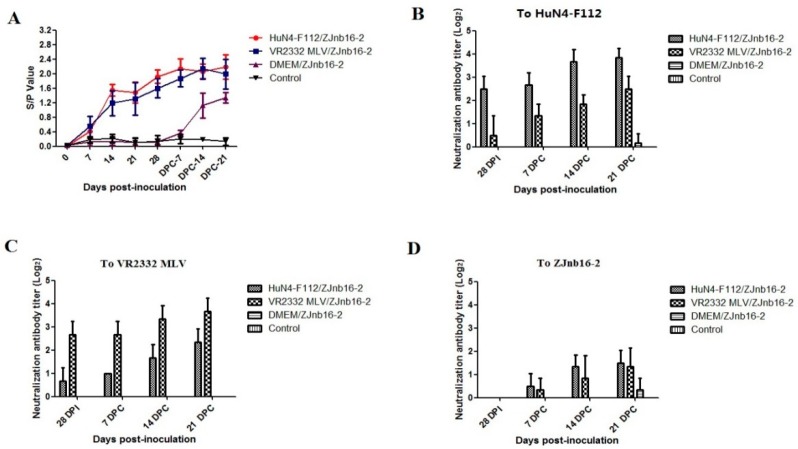
Detection of porcine reproductive and respiratory syndrome virus (PRRSV)-specific N protein antibody and neutralization antibody in each group. (**A**) N protein antibody level after vaccination and data are expressed as mean ± SD (error bars); Neutralization antibody titers against HuN4-F112 (**B**), VR2332 MLV (**C**), and ZJnb16-2 (**D**) of serum collected at 28 days post vaccination (DPV), 7 DPC, 14 DPC, and 21 DPC.

**Figure 4 pathogens-09-00059-f004:**
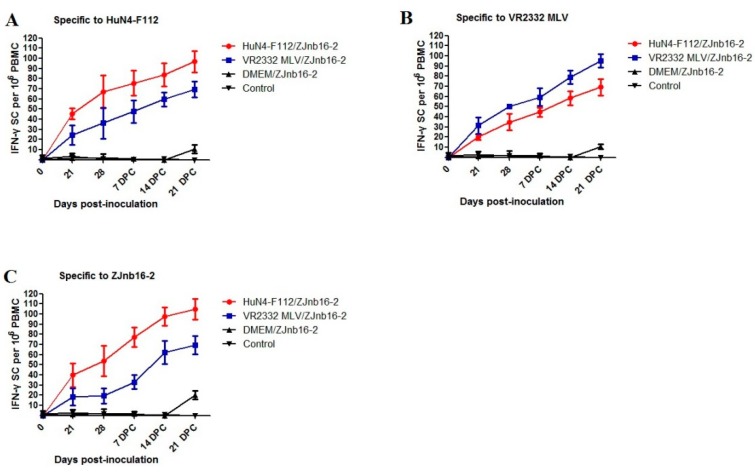
Kinetics of IFN-γsecreting cell responses against in vaccinated piglets. (**A**) The number of interferon (IFN)-γsecreting cells (SC) against strain HuN4-F112; (**B**) the number of interferon (IFN)-γSC against strain VR2332 MLV; (**C**) the number of interferon (IFN)-γSC against strain ZJnb16-2.

**Figure 5 pathogens-09-00059-f005:**
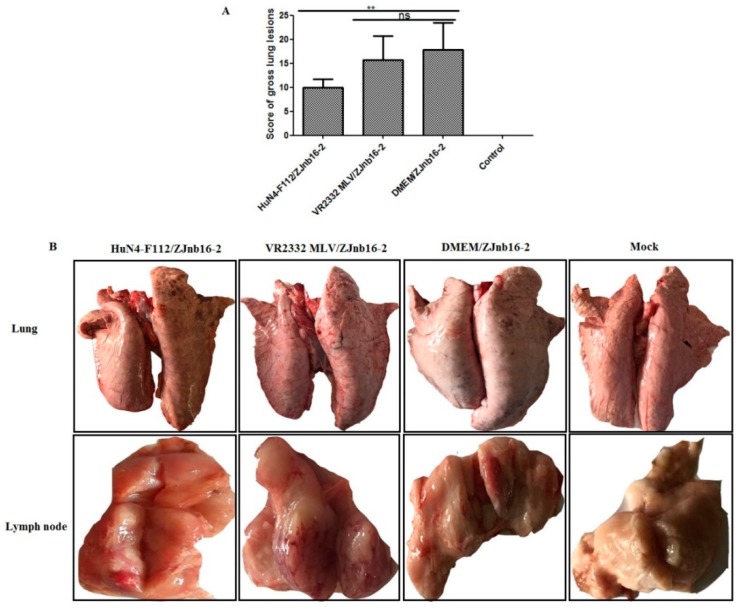
Lung and lymph nodes gross pathological examination of the challenged piglets. (**A**) Scores of gross lung lesions were shown as mean ± standard error (error bars), which were graded based on percent lung area affected; (**B**) Visible pathological lesions in lung and lymph nodes. ** *p* < 0.01.

**Figure 6 pathogens-09-00059-f006:**
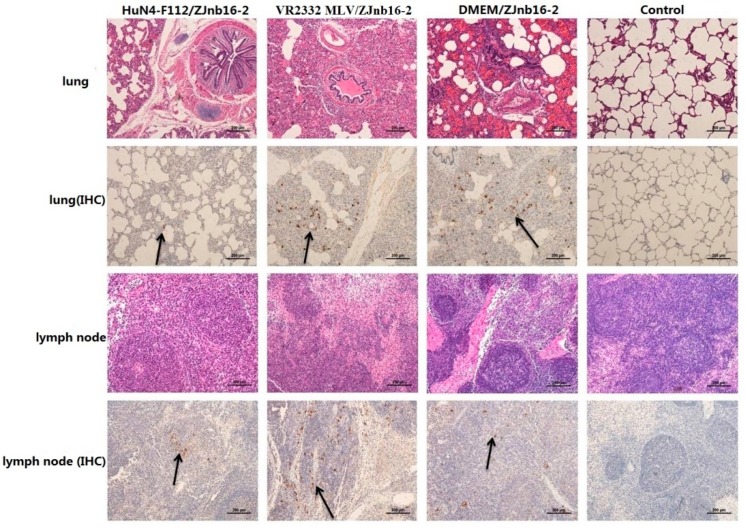
Pathologic and immunohistochemical examination in lung and lymph nodes of each group.

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
