# Peer review of "Efficacy Evaluation of Two Commercial Vaccines Against a Recombinant PRRSV2 Strain ZJnb16-2 From Lineage 8 and 3 in China"

_pathogens, 2020, doi:10.3390/pathogens9010059_

Round 1
Reviewer 1 Report
The manuscript “Efficacy evaluation of two commercial vaccines against a recombinant type 2 PRRSV strain ZJnb16-2 from lineage 8 and 3 in China” by Han et al. describes the results of an efficacy study performed using two different MLV PRRSV vaccines against a single heterologous challenge. The study is a classical efficacy study in which young piglets were vaccinated and four weeks later exposed to a fully virulent PRRSV isolate. The novelty of the study is not high but it could be of interest for people working in this field. The experimental design is, generally speaking, appropriate, although some modifications should be done before the manuscript can be considered suitable for publication. In addition, there are other aspects in the manuscript that should be reviewed.
In relation to the experimental design, the aspects that should be clarified are the following:
The authors present the data of the titer of neutralizing antibodies (NAs) in each group of pigs on day 28 p.v. (i.e. the day of challenge) but they do not monitor the evolution of the titer of NAs after challenge against the challenge strain. This information is very relevant because, even though no challenge strain-specific NAs could be detected the day of challenge, it is possible that the immune system have been primed by vaccination and that NAs appeared shortly after challenge, contributing to explain the reduction on viremia and lung pathology, as well as the improvement in productive parameters (i.e. ADWG) found in partially protected pigs. This information is very important and should be provided because the authors indicate that no NAs were detected and that they did not contribute to protection. However, as protection was not characterized by sterilizing immunity (i.e. complete prevention of infection) but rather by shorter viremia and improvement in clinical signs, lesions and productive parameters, the role of NAs cannot be completely ruled out with the current data. Thus, data on evolution of the titer of NAs against vaccination and challenge viruses after challenge need to be provided. The same commentary applies to cell-mediated immunity measured by frequency of PRRSV-specific IFN-gamma secreting cells. Although they can be detected before challenge, their numbers are relatively low and it is important to know how they have evolved upon challenge. These data should be provided. Although in the graphics statistically significant differences are represented, in the corresponding heading in Material and Methods, the authors only indicate that differences in ADWG and virus titers were analyzed using one-way or two-way analysis of variance. However, it is not indicated how differences in rectal temperatures, clinical signs and lung pathology were evaluated. It is likely that individual variability in these parameters prevent the use of parametrical tests. How these parameters were statistically evaluated? In addition, as the study comprised four groups, after ANOVA a post-test should have been applied to compare groups two by two. This information is lacking and should be provided. In relation to the technique used to evaluate the presence of NAs the authors use porcine alveolar macrophage (PAM) cultures and indicate that the titers are estimated using the method of Reed and Muench. On one hand, although cultures of PAM can be used to determine the titer of NAs against PRRSV, the technique is normally performed using viruses adapted to grow in MARC-145 because the titers of NAs obtained in PAM cultures are usually very low. The titers generally used as predictors of protection have been obtained in seroneutralization assays (SN) performed in MARC-145 (Osorio et al. Virology, 2002, 302: 9-20; López et al. Clin Vaccine Immunol, 2007, 14: 269-275) so the use of this cell line would have helped to interpret the results. Besides, vaccine viruses used to immunize the pigs have been adapted to grow in MARC-145 cell line by a number of serial passages and they tend to grow poorly in PAMs. Thus, these viruses usually have a much lower titer in PAMs than in MARC-145. Consequently, for the vaccine viruses, the NA titers might have been underestimated because the real amount of virus to which the sera were confronted was much higher than estimated. The authors should explain why they selected PAM cultures to determine the presence of NAs in the sera of vaccinated pigs, particularly against vaccine viruses which are grown in MARC-145 cells or other derivates of MA-104 cell line. In addition, viral titers are usually estimated as the highest serum dilution that reduce at least by 90% the number of fluorescent foci when the technique is revealed by immunofluorescence assays. The method of Reed and Muench is frequently used to determine virus titer but not to determine antibody titers. The authors should explain why they have chosen this method which is not comparable to other studies in the literature and, if possible, indicate the titer using a conventional assay. In any case both (i.e. the use of PAM cells, particularly with vaccine viruses, and the use of Reed and Muench method to calculate the titers) make comparisons with other studies impossible to perform and make difficult to evaluate the real response of the pigs. In the experimental design the recording of clinical signs is indicated and a figure is provided indicating summation by group and days and differences between groups. However, the clinical score used to evaluate the clinical status of the animals is not provided in the manuscript. This clinical score evaluation system should be provided.Besides the abovementioned issues there are many other details along the text that need to be modified to improve the quality of the manuscript. Among them, the most relevant are the following:
Abstract lines 20-21. It is stated that HuN4-F112 vaccine provided partial protection while the efficacy of VR2332 MLV was limited. “Partial protection” and “limited protection” are very subjective concepts. In fact, the protection provided by VR2332 could be considered partial. The sentence should be rewritten indicating that both vaccines provided partial protection although of efficacy of HuN4-F112 was better for most of the parameters analyzed. In the same way, in lines 23-24 it is indicated that none of the vaccines provided full protection in relation to lung pathology. Introduction line 36. It is stated that PRRSV genome encodes at least 12 ORFs. However the references provided do not support this statement. Even more, in the last edition of Diseases of Swine it is indicated that “The genomic organization of PRRSV is similar to that of other arteriviruses, consisting of approximately 15,000 nucleotides organized into about 11 ORFs (1a, 1b, 2a, 2b, 3, 4, 5a, 5, 6, 7 and a short transframe ORF) that are expressed from genomic and subgenomic (sg) mRNAs (sgmRNAs) (Zimmerman et al. 686-687 in: Zimmerman, Jeffrey J., Locke Karriker, Alejandro Ramirez, Kent Schwartz, Gregory Stevenson. Diseases of Swine, 11th Edition. Wiley Professional, Reference & Trade). Please, correct. Introduction lines 28-39. It is indicated that PRRSV is divided into two genotypes. However, the International Committee of Virus Taxonomy in the last release divides PRRSV into two species and so it is recognized in the last edition of “Diseases of Swine”: “PRRSV type 1 (PRRSV‐1) and PRRSV type 2 (PRRSV‐2) are currently regarded as two species classified with 15 other species of primate, rodent, and equine viruses in the family Arteriviridae (Zimmerman et al., 685 in: Zimmerman, Jeffrey J., Locke Karriker, Alejandro Ramirez, Kent Schwartz, Gregory Stevenson. Diseases of Swine, 11th Edition. Wiley Professional, Reference & Trade). Please, correct. Introduction lines 39-40. It is stated that PRRSV-2 is divided into 9 lineages. This is true but the right reference for this is Shi M, Lam TT, Hon CC, Hui RK, Faaberg KS, Wennblom T, Murtaugh MP, Stadejek T, Leung FC. Molecular epidemiology of PRRSV: A phylogenetic perspective Virus Research 154 (2010) 7–17. In the reference cited by the authors, this system is used to classify PRRSV from China, but the classification system is not described. Please, refer to the right reference. Introduction lines 51 and 52. The authors state that commercial vaccines may not provide sufficient protection against the risk of new outbreaks of PRRSV of lineage 3. However, they previously talk about recombinant viruses which, be definition, might belong to lineage 3 or to a different lineage, depending on the viruses that participate in the recombination process and the location of the recombination effect in the PRRSV genome. This should be better explained. Even more, first paragraph in discussion (lines 123-133) should be moved to introduction because this will help to explain the situation in China and the objective of the study and it is not necessary in Discussion. In addition Table 1 should be deleted because it does not contain any data obtained in the study. Nonetheless, a description of the viruses circulating in China could be added, if the authors wish to do it. Introduction line 53. Where it says clinical symptoms it should say clinical signs. Please replace symptoms by signs along the whole text because this is the right expression to refer to clinical observations in animals. Introduction line 53. Please remove “viraemia” of the list of clinical signs because viraemia is not a clinical sign. Introduction line 64. Where it says modified-live vaccines should say modified-live virus vaccines (MLV stands for Modified Live Virus). Please, correct. The objective should be clearly stated and described. Please indicate the lineage of the vaccines and the challenge virus as well. Subheadings in Result section should be neutral and indicate the parameter for which the results are going to be presented as it is subheading 2.1. On the contrary the name of some of the other subheadings evaluate the results instead of describe them aseptically as it should be done in Result section (e.g. 2.3 No cross neutralization reactivity against ZJnb16-2 in vaccinated piglets). In results, please clearly state where differences are statistically significant and where the differences are only numerical. In addition, please, indicate the level of significance (i.e. P<X). Figure 1D could be deleted because only one pig from one experimental group was dead and it is not necessary. Results, line 84. It is stated that viremia was detected only in one pig upon vaccination. Was viremia checked in PAM or MARC-145 cells? Viremia upon vaccination should be checked in MARC-145 because vaccine viruses replicate better in this cell line than in PAM cells. PAM cultures are not sensitive enough. Please, provide the data in MARC-145 cell line. Results, line 99 refers to Figures 4A and 4B. However, the data of unvaccinated and unchallenged groups are not depicted in these Figures. Was the number of PRRSV specific IFN-gamma secreting cells studied? The data should be obtained and depicted in the Figures. Results, line 107. The right term in veterinary science is necropsy. Autopsy refers only to humans (it comes from the Greek word “autós”, meaning oneself). Discussion, lines 135-137. This sentence should be deleted because it is not accurate. The authors could indicate that, although in both cases only partial protection was achieved, HuN4-F112 seemed to be more effective in controlling the negative consequences of challenge. Discussion, line 143. It is indicated that ADWG in similar in pigs vaccinated with HuN4-F112 than in the controls, but there is a negative effect on growing during the first week post-challenge. This should be clearly stated. Discussion, lines 155-157. The sentence “Generation of broadly neutralization antibody…..” makes no sense. In fact, it is the other way around. The exposure to multiple heterologous strains along time is considered to trigger the appearance of broadly neutralizing antibodies but the presence of broadly neutralizing antibodies is not related to prolonged infections nor to heterologous infections. Please, rephrase the sentence to express the idea clearly. Discussion lines 169-170. The sentence “the improved efficacy of HuN4-F112 vaccine against a genetic related PRRSV was mainly resulted from the stronger cell-mediated immunity response” should be rephrased. The specific immune response was measured before challenge but not at any time after challenge. However, the protective effect reported in this study was only observed from Day 14 post-challenge onwards because on Day 7 most parameters were alike between groups. Still, there is no information about the NAs or the number of IFN-gamma secreting cells at that time point upon challenge. Consequently, it can only be suggested that the cellular immune response elicited by vaccination with the different vaccine strains used in this study might have played a role in the protection achieved. On the contrary, it is not possible to completely rule out the role of NAs. Discussion, lines 178-182. These sentences refer to experimental approaches to improve vaccine efficacy, but they are not related to the subject of this study nor to the results of the study. Consequently they should be removed unless it is indicated that the current vaccines are not adequate to induce sufficient protection and new approaches are necessary to improve the immunogenicity of PRRSV vaccines. Materials and Methods, subheading Cell, Vaccines and Virus. Please, indicate the lineage to which each of the viruses used in this study belongs. Materials and Methods, line 192. PAM stands for “Porcine Alveolar Macrophage”. Although the term Pulmonary Alveolar Macrophage is occasionally used, alveolar macrophage is the correct way to refer to the macrophages that are free in the alveoli, in close contact with the type I and II epithelial cells of alveoli, but separated from the wall. These macrophages are different of other pulmonary macrophages, as interstitial macrophages, which reside in the parenchyma between the microvascular endothelium and alveolar epithelium or intravascular macrophages, which are found apposed to the pulmonary capillary endothelium in different species, including pigs. The cells used to grow PRRSV in vitro, are porcine alveolar macrophages. Materials and Methods, subheading Animal experiments and Clinical assessment. Please, indicate when and how pigs are vaccinated. Materials and Methods, subheading Virus titration should be named Viremia and virus titration because the data of viremia are explained in this subheading. Materials and Methods, subheading Virus titration. Please, indicate number of PAMs/well and what means 1% antibiotic-antimycotic (i.e. indicate which antibiotics and which antimycotic and the actual concentration of each of them). Materials and Methods, line 222. Please, refer properly the method of Reed and Muench to estimate viral titers (Reed JJ, Muench TH. (1938). A simple method to estimating fifty percent end points. Am. J. Hyg. 27, 493-497). Materials and Methods, subheading Assessment of PRRSV antibody. Please, indicate the number of PAMs/well used in the SN assay. Materials and Methods, line 249. Please, refer to the right reference to describe the scoring system for lung lesions (Halbur PG, Paul PS, Frey ML, Landgraf J, Eernisse K, Meng XJ, Lum MA, Andrews JJ, Rathje JA. (1995) Comparison of the pathogenicity of two US porcine reproductive and respiratory syndrome virus isolates with that of the Lelystad virus. Vet Pathol 32: 648–660). The paper by Li et al (2014) referred in the manuscript only use the method previously described by Halbur et al. Did the authors determine the cause of death in the dead pig in the group vaccinated with MLV VR2332? Please, provide information regarding the method followed to determine the cause of death of that pig and to rule out other causes of death. The photographs in Figure 5 do not have sufficient quality. Please, improve the quality of the photographs. Do the lung and lymph node represented in group MLV/ZJnv16-2 correspond to the dead pig? The lung and lymph node appear to be autolytic. Please, make sure that you provide photographs of pigs euthanized under the same conditions. References. Reference 47 should be deleted. The use of English should be reviewed, and improved, along the manuscript.All these points should be solve before the article can be considered suitable for publication.
Reviewer 2 Report
The manuscript describes one laboratory efficacy study in which two comercial vaccines are tested in front of a challenge with a recombinant PRRSV-2 lineage 3 strain.
The experimental design, methods used and results obtained are quite clearly exposed and are adequate. However, discussion is in many instances a repetition of the results, and the strict discussion should be further expanded.
One important deficiency has been identified in the statistical analysis used. Due to the fact that the different groups were kept separately in different isolation rooms, the experimental unit became the room and not the pig. Since the experimental unit (room) was not replicated, analysis of results is not posible. Then, the statistical analysis should be replaced by descriptive statistics:
The experimental unit, according to the Journal of Animal Science (and numerous text books) is defined as the smallest unit to which a treatment is imposed. The experimental unit is determined by how the treatments are randomized and how the animals are housed once treatments have been randomized.
https://dl.sciencesocieties.org/files/publications/jas/jas-instructions-to-authors-110617.pdf
EFSA published The Guidance on Statistical Reporting (2014). https://efsa.onlinelibrary.wiley.com/doi/epdf/10.2903/j.efsa.2014.3908
VICH GL9 (GCP) has multiple sections in which the experimental unit is mentioned (6.3.8 and 6.3.17). While a definition is not given, a differentiation is made between animals and experimental units (not always one and the same).
https://www.ema.europa.eu/en/documents/scientific-guideline/vich-gl9-good-clinical-practices-step-7_en.pdf
The CVMP guideline on statistical principles for clinical trials for veterinary medicinal products makes clear the difference in the animal as the experimental unit vs some other group of animals as the experimental unit.
https://www.ema.europa.eu/en/documents/scientific-guideline/guideline-statistical-principles-clinical-trials-veterinary-medicinal-products-pharmaceuticals_en.pdf
If treatments are randomly assigned to rooms, then the room is the experimental unit even though the treatment might be administered on an animal to animal basis. It does not matter if each individual animal has measurements taken on it such as weighing each animal. If all animals within that room receive the same treatment, the room is the experimental unit. If those experimental units are not replicated, due to statistical theory, a statistical analysis is not possible due to the lack of the appropriate number of degrees of freedom. Applying (incorrect) p-values to non-replicated data can be deceiving and lead to incorrect decisions no matter the kind of study it is.
The statistical power of the study, having only 6 pigs per treatment, should be provided and used for the interpretation of the results, which could be misleading.
Other minor comments are provided below:
The authors should review the nomenclature of PRRSV, since the priorly called type 1 and type 2 are now considered two different species, PRRSV-1 and PRRSV-2. Abbreviations are used throughout the document. However, in some instances the full name is not always provided the first time that the abbreviation is used. Please review accordingly. Provide some background about QYYZ-like and HP strains before they are mentioned for the first time. Otherwise, the context is missing. In line 110, "severe lung pathological lesions" must be described. Please provide more information about the source of the pigs: did they come from a commercial farm? Which are the characteristics of the source farm? Plase explain how the ADE (antibody dependent enhancement) issue was sorted out to establish a PRRSV virus neutralization assay in PAM.
Reviewer 3 Report
This study makes an important contribution in terms of vaccine efficacy that will be of considerable practical use to pig farmers and scientists. The results show clearly that one vaccine is superior to the other in terms of clinical score, several immunological metrics, and to a lesser extent virus titer reduction. The scientific content and the style of writing are both appropriate for this journal.
Minor changes
Fig. 1c, 2b, 2c, 5a - please clarify in the statistical methods section how you handled multiple comparisons.
Fig. 1d - we need different symbols or some other way to see when lines are stacked, either on each other or on an axis - uninfected and the unvaccinated controls are not visible to my eye, even when fully zoomed-in.
Fig. 1 legend or the figure and any others where multiple replicates are done should specify the number of piglets in each treatment group.
Fig. 3b viruse --> virus
It would be a little easier to parse if the type 5 live vaccine were referred to in figures and legends as VR2332 rather than MLV (which looks at first like a number of other virus names to me).
Round 2
Reviewer 2 Report
The authors did not understand the comments about the statistical analysis. It does not matter that individual measures were taken from each pig. As long as the groups were housed separated from each other, the experimental unit becomes the room and not the pig. If the animals within each group were housed separately, then the experimental unit would be the pig, but that was not the case according to the description in the manuscript. Consequently, the data could only be used for descriptive statistics, and no p value could be calculated.
